# How Energy Consumption and Pollutant Emissions Affect the Disparity of Public Health in Countries with High Fossil Energy Consumption

**DOI:** 10.3390/ijerph16234678

**Published:** 2019-11-24

**Authors:** Xinpeng Xing, Jianhua Wang, Tiansen Liu, He Liu, Yue Zhu

**Affiliations:** 1School of Business, Jiangnan University, Wuxi 214122, China; xinpeng@jiangnan.edu.cn (X.X.); jianhua.w@jiangnan.edu.cn (J.W.); 2School of Economics and Management, Harbin Engineering University, Harbin 150001, China; 3School of Management, Harbin Institute of Technology, Harbin 150001, China; zhuyue@hit.edu.cn

**Keywords:** energy consumption, pollutant emissions, public health, healthcare resources, highly energy-consuming countries

## Abstract

Public health issues are a global focus, but recent research on the links between fossil energy consumption, pollutant emissions, and public health in different regions have presented inconsistent conclusions. In order to quantify the effect of fossil energy use and pollutant emissions on public health from the global perspective, this paper investigates 33 countries with high GDP and fossil energy consumption from 1995 to 2015 using a fixed effect model. Further, this paper utilizes heterogeneity analysis to characterize the disparity of countries with different features. Empirical results indicate that total fossil energy consumption is beneficial to the life expectancy of the population (LEP), but pollutant emissions (PM10 concentration and greenhouse gas scale) have a negative effect on LEP. Moreover, the heterogeneity test indicates that pollutant emissions lowers LEP in net energy importers more than in net energy exporters, and the effect of such emissions in low- and middle-income countries on public health is more harmful than that in high-income countries. These findings suggest that it is a greater priority for governments to strengthen the control of pollutant emissions through enhancing the efficiency of energy consumption, rather than by reducing its scale of use in low- and middle-income, and net energy importing countries. Additionally, governments also need to focus on the volatility of pollutant emissions in high-income countries with necessary control measures.

## 1. Introduction

In recent years, climate change and deterioration of environmental quality have attracted an increased global focus relating to environmental governance, energy restructuring, and the impact of pollutant emissions on public health. Rapid economic growth not only triggers massive use of fossil energy, but also leads to a rise of emissions of various types of pollutants [1]. Pollutant emissions caused by fossil energy consumption have been analyzed to endanger public physical and mental health [2,3], and numerous previous studies have found that such pollutant (e.g., PM10, PM2.5, and SO_2_) can have a negative effect on public health [4,5,6]. Sharma et al. [7] examined the link between air quality and public health in the Delhi region, and found that pollutant emissions were triggering an increase in diseases in industrialized economies. Khan [8] revealed that in the United States and Denmark, environmental pollution can increase the risk of psychiatric disorders, thereby realizing major threats to public health. Kelly and Fussell [9] organized an empirical test in terms of the link between air pollution and public health in Netherlands, and found that air pollution not only exerts a greater effect on established health end-points, but is also associated with broader numbers of disease outcomes. The results of Perera [10] indicated that pollutant emissions and climate change caused by fossil fuel use produce multiple threats to children’s health. Xu et al. [1] revealed that the effect of pollutant emissions on public health was obvious in almost all regions in China. However, a few studies found that, in some regions, there is no a significant relationship link between pollutant emissions and public health. For instance, Torres et al. [11] explored such links in Portugal, finding that gradual improvement of air quality did not lead to the fall of the mortality rate in Alentejo and Algarve. In order to clearly address these unresolved issues and disputes, this paper classifies recent studies in terms of the link between pollutant emissions and public health, as shown in Table 1.

Table 1 presents the key areas and main conclusions of existing studies. As shown in Table 1, most previous studies empirically investigated one country, but a few studies have recognized that the effect of air quality on public health is affected by inter-country heterogeneity in income, social welfare, public health expenditure, etc. Accordingly, in addition to the controversy over the link between pollutant emissions and public health, the disparity of the effect of pollutant emissions on public health is another research gap that needs to be explored regarding differences in income, social welfare, and public health expenditure among different regions or countries.

Another topic of concern is the effect of fossil energy consumption on public health. However, the findings on the link between fossil energy consumption and public health are also controversial. Several previous studies also confirmed that as one of major sources of pollutant emissions, fossil energy consumption threatens public health, and its negative effects are even more significant in areas where energy consumption is concentrated [5,18,19]. Therefore, fossil energy and the pollutant emissions caused by fossil energy consumption are regarded as determinants of public health [20,21,22]. However, some previous studies for specific regions and countries showed that fossil energy consumption has a negligible impact on public health, and that even in industrialized countries the social benefits created by energy consumption have been shown to spread population health risks [23,24,25]. For example, the results of [15] showed that fossil energy, especially coal, has a positive impact on Nigeria’s economic growth rate. Rauf et al. [26] and Martins et al. [21] believed that in industrial counties, fossil energy consumption could push forward the improvement of economic factors, for example, the employment rate, residents’ income, market activity, tax scale, and international trading system, in the process of improving public health [21,26]. Therefore, the social welfare created by fossil energy consumption directly results in a more favorable human living environment and healthcare system, thereby contributing to better public health [1,27]. In order to analyze the outstanding problems and disputes more effectively, this paper collates the relevant studies on fossil energy and public health (shown in Table 2).

As presented in Table 2, it is clear that previous studies on fossil energy consumption and public health were also focused on a particular country or region. The existing studies have not reached a consensus on the linkage between fossil energy consumption and public health. Furthermore, for countries with different characteristics, income level, medical and health resources, and energy structure may lead to significant differences in the link between environmental pollution and public health [27]. Overall, the positive and negative effects of energy consumption on public health have been explored. However, existing studies are still unable to determine the direction and extent of the impact of such consumption on public health, and whether the impact will show significant differences in different regions or countries.

By analyzing the existing literature on the link between fossil energy consumption and public health, as well as the link between pollutant emissions and public health, there are still some unresolved problems. First, the link between pollutant emissions and public health and the link between fossil energy consumption and public health are still controversial, and need to be further explored. Second, previous studies focused on a country or some regions of a country, and few studies conduct empirical analysis of the link between fossil energy consumption, pollutant emissions, and public health in different countries around the world. Therefore, the link between energy consumption, pollutant emissions, and public health needs to be further discussed, including whether it is affected by the disparity in income, social welfare, and public health expenditure among regions or countries.

In order to fill prior research gaps, this paper strives to explore the impact of environmental pollution and fossil energy consumption on public health via investigating 33 countries whose average proportion of fossil energy consumption was higher than 70% during 1995 to 2015, and whose GDP ranked in the top 50 in the world in 2015, taking into account social factors. Furthermore, considering the national heterogeneity, this paper aims to explore the disparity of the effect of pollutant emissions and fossil energy consumption on public health in net energy importers and exporters, as well as low-, middle-, and high-income countries.

The main contributions of this paper are as follows: First, this paper extends the environment, energy, and public health literature to the global level by exploring the impact of fossil energy consumption and air quality on public health from a broader perspective. Second, a range of economic and social factors, such as social resource allocation, healthcare resources, and population distribution, are also considered to test the impact of fossil energy consumption and pollutant emissions on public health expenditure. Third, this paper identifies the disparity of the effect of pollutant emissions and fossil energy consumption on public health by examining the links between three variables regarding heterogeneity of 33 countries.

The remainder of this paper is organized as follows: The research methodology is portrayed in Section 2. The empirical results are stated in Section 3. The heterogeneous test is organized to visualize the disparity in countries with different features in Section 4. We present a discussion and some implications in Section 5. Finally, Section 6 concludes the study, and clarifies limitations and directions for future research.

## 2. Research Design

### 2.1. Data Collection

Considering the integrity and availability of all data, this paper organizes the time series interval between 1995 and 2015. Our data were collected from the World Bank Database and Economy Prediction System (EPS) Global Statistics Database developed by Beijing Forecast Information Technology Co., Ltd., Beijing, China. Specifically, the concentration of PM10 emissions is collected from the EPS Database, with all other indicators from the World Bank Database. Determination of the investigated countries was based on the following principles. In view of the close relation between energy consumption and GDP, this paper limits these countries to those whose GDP ranked in the top 50 in the world in 2015 and with an average proportion of fossil energy consumption in 1995–2015 of no less than 70%. Accordingly, we analyzed the following 33 countries: USA, China, Japan, Germany, UK, Italy, Canada, Korea, Russia, Australia, Spain, Mexico, Netherlands, Turkey, Saudi Arabia, Argentina, Poland, Belgium, Thailand, Austria, Egypt, South Africa, Denmark, Malaysia, Singapore, Colombia, Ireland, Chile, Portugal, Greece, Peru, Czech Republic, and Kazakhstan, as shown in Figure 1. 

With respect to the selection of our samples, some additional notes are as follows. First, although the International Monetary Fund (IMF) has long predicted that the GDP of Iran and Venezuela would rank among the top 50 countries in the world, the World Bank did not publish the specific GDP values of these two countries in 2015. Thus, this paper does not analyze these countries but investigates whether countries whose GDP ranked 51 and 52 in 2015 are in line with our criteria for sample selection. Second, as data were missing for several indicators, some countries (or regions) within the top 50 global GDP ranking, including the United Arab Emirates, Hong Kong, China, and Israel, were not considered in the investigation.

Through calculations, it was found that from 1995 to 2015, these countries’ total population accounted for 44.31% of that of the world; thus, our sample presents the prominent contradiction among energy consumption, pollutant emissions, and public health in the process of global industrialization, and accordingly provides robust evidence for improving the overall state globally. Overall, this study used 13 variables, and collected 693 sets of observations and 9009 data points. It should be noted that there were a few missing data points in terms of some variables in 2014 and 2015; these data were estimated based on the distribution trend of data in previous years.

### 2.2. Variables

Dependent variable: This study used “life expectancy of population (LEP)” as the dependent variable to describe the level of public health. As LEP can reflect the overall health level of a country’s residents, this paper argues that it will impartially present the effect of energy consumption on public health and thus avoid exaggerating the negative consequences of this consumption [30,31,32].

Independent variables: A considerable number of factors can help to quantify the level of energy consumption and pollutant emissions in one country. In view of their close relationship with actual production behavior and demographic factors, this study used “energy use per capita (EUPC)” and “GDP per unit of energy use (GDPEU)” as the first set of independent variables that describes the consumption scale and consumption efficiency in one country. Further, in view of the different forms of major pollutant emissions caused by energy use, “Green House Gas (GHG) emissions per capita (including CO_2_, methane, HFC, PFC, and SF_6_) (GHGPC)” and “emission concentration of inhalable particulate matter (PM10)” were designed to be another set of independent variables to describe the state of pollutant emissions. There is no global consensus on whether GHGs are pollutants. The United States Supreme Court ruled CO_2_ to be a kind of air pollutant because, although the stability of environmental effect of CO_2_ is weak, its source is increasing and it has been found to constantly have negative effects on the natural environment [33,34]. Additionally, considering the high proportion of fossil energy consumption in our sample countries, GHGs may have a great impact on many factors in one country; thus, this paper views GHGs as a kind of pollutant [35].

Control variables: This study designed control variables based on following principles. First, these variables have been proved to significantly affect the level of dependent and independent variables, as well as their links. Second, since the scale of a country’s economy will determine its mode of energy consumption via a series of related activities, this study also tested the heterogeneity between high-income and low- and middle-income countries within our samples to present the wide-ranging effect caused by the disparity in national income. According to the above two principles, this study designed three types of control variables. The first set was the population distribution. Previous studies supported that the urban scale and concentration of the urban population are usually positively correlated with energy consumption demands, and the mitigation of climate change will contribute more to improving the public health in cities than that in non-urban areas [18,36]. This implies that the negative effect of energy consumption on public health may only appear in cities rather than nationally. Accordingly, this study used “population density (PD)” and “population in urban agglomerations of more than 1 million (% of total population) (PUA)” to describe population distribution in one country. The second set was the allocation of social resources. Previous studies have confirmed that the equality of social resource allocation usually positively affects LEP, and resources regarding the employment and education are even residents’ basic rights [37,38]. Accordingly, this study used “unemployment, total (% of total labor force) (UEM)” and “education expenditure (% of Gross National Income) (EE)” to describe the state of social resource allocation. The third set was the scale of medical and healthcare resources. Previous studies on global health strategies showed that irrespective of the triggers of regional diseases, the healthcare system is always a fundamental determinant of public health, including government investment for residents’ health and residents’ willingness to maintain their own health [39]. Several studies predict the gradually blurred boundary of health risks from climate change and pollutant emissions, indicating that the effect of such risk in rich and poor regions will narrow, thus increasing the actual demand for high-quality healthcare systems in the future [40]. Accordingly, this study used “improved healthcare facilities (IHF)”, “the proportion of public health expenditure to GDP (PHE)”, and “the proportion of per capita health expenditure to per capita GDP (PCHE)” to describe the state of healthcare resources.

### 2.3. Measuring Models

This study used a fixed effect model to organize the empirical test based on the following considerations. First, we investigated the overall state of sample countries, finding a number of factors that do not experience significant volatility over time, but which may affect the level of dependent and independent variables, as well as their links. Second, empirical results will be more appropriate to present the relationship among variables within our sample countries rather than other countries. The Hausman test verifies that the *p*-value of each model is less than 0.01, which further indicates that the fixed effect model is better than the random effect or mixed effect models. The regression model is defined as follows:(1)$$Y_{it}=\alpha +\gamma \times C_{it}\times \beta \times X_{1it}+state_{i}+year_{t}+\varepsilon_{it}$$
(2)$$Y_{it}=\alpha +\gamma \times C_{it}+\beta \times X_{2it}+state_{i}+year_{t}+\varepsilon_{it}$$
where $Y_{it}$ indicates the dependent variable, and $X_{1it}$ and $X_{2it}$ indicate the energy consumption and pollutant emissions, respectively. $C_{it}$ indicates the control variable, and γ and β indicate regression coefficients. Further, $state_{t}$ is the fixed effect at the national level used to control the factors that do not alter significantly over time, e.g., climatic conditions and ecological resources (forest coverage). $year_{t}$ is the fixed effect at the time level used to control the macro-factors affecting economic and social activities in one country in a given year, e.g., polity and industrial policies.

It should be noted is that this paper agrees with the use in previous studies of time-lag variables that help to describe the lag effect of numerical volatility of variables. However, the time lag of variables was not considered in this paper for the following two reasons. First, with respect to LEP, both the World Bank Database and China Statistical Yearbook published the previous year’s data in the following year. Accordingly, this paper argues that these published data have fully considered the factors that may affect LEP, and thus evaluate its numerical value. Second, some factors may moderate the impact of energy consumption and public health; thus, this paper does not contain variables with significant lag effect, but presents the transient effects of control and independent variables. 

The descriptive statistics shown in Table 3 indicate that the numerical value of variables varies among countries, which suggests that the volatility of public health is the result of the joint effect of multiple factors.

## 3. Empirical Test

### 3.1. Correlation Analysis

Due to limited space, this paper only explains some important sets of correlation coefficients (Pearson coefficients) to demonstrate the overall correlation of these variables in our sample countries. First, there is a highly positive correlation between PD and PUA, with a high correlation coefficient (0.623, *p* ≤ 0.01), which indicates that countries with higher PD are more likely to form population agglomerations, thereby leading to the difference in the level of public health between metropolises and other regions. Second, variables regarding healthcare resources are significantly positively correlated with each other (the correlation coefficients between IHF and PHE, PHE and PCHE, and PCHE and IHF are 0.453, 0.318, and 0.841, respectively; *p* ≤ 0.01). This indicates that better national healthcare systems (including hardware facilities and institutional guarantees) correlate with higher levels of residents’ health, and the scale of PCHE may be greater than that in countries with undeveloped healthcare. Third, control variables are closely correlated with independent and dependent variables as a whole, including positive and negative correlations. Fourth, there is a high correlation between each pair of independent variables, which indicates that higher EUPC and lower consumption efficiency will both trigger more serious pollutant emissions. Fifth, each independent variable is significantly correlated to LEP, with correlation coefficients of 0.339, 0.355, 0.148, and −0.406 following the order of independent variables in Table 1, respectively (*p* ≤ 0.01). However, it should be noted that PM10 emission concentration is significantly negatively correlated with LEP, but EUPC is positively correlated with LEP (0.339; *p* ≤ 0.01). This case corresponds to a previously discussed assumption that the factor impeding the improvement in LEP may be undeveloped production technologies rather than energy consumption itself.

### 3.2. Regression Analysis and Robustness Test

The robustness test aims to verify whether empirical results will considerably change with the parameters and the rationality of variables designed in this paper. Following Wu et al. [26], we consider the general rule of numerical volatility of some variables within a period of time. For instance, the average annual use of energy presented a steady upward trend since 2000 in China, but fell sharply after a period of fluctuation from 1995 to 2015 in Japan. This difference may be caused by the joint effects of policy regulation and residents’ behavior patterns. Accordingly, this study used the time interaction effect of control and independent variables to verify whether this interaction changed the impact of control and independent variables on LEP. Table 4 presents the empirical results based on the fixed effect model, and Table 5 records the time interaction effect to verify the robustness of variables’ function.

Through comparing Table 4 and Table 5, it can be found that both direction and significance of regression coefficients of each variable are consistent no matter whether considering the effect of time, which supports the robust contribution of control and independent variables at the time level. Overall, the value of adjusted R^2^ indicates that the volatility of LEP can be well explained. Additionally, there is smaller clustering standard error of control and independent variables at the national level. Further, this paper more broadly discusses the empirical results.

First, EUPC (energy use scale) significantly contributes to the increase of LEP, whereas the effect of GDPEU (energy use efficiency) is more positive. Further, both GHGPC and PM10 negatively affect LEP, with the latter playing a statistically significant damaging effect. It indicates that solid particulate matter may be more harmful to public health than GHG in industrialized countries. Although regression coefficients of EUPC and PM10 are small, their contribution should not be considered weak in terms of their measuring units. Theoretically, the common state in the sample countries is that when EUPC increases by 1 kg of oil equivalent or PM10 emission concentrations increase by 1 microgram/m^3^, LEP will increase by 0.0016 years and decrease by 0.0671 years, respectively. However, for GDPEU, an increase in the output of 1 PPP $ per kg of oil equivalent will boost LEP by 0.7273 years. In spite of the theoretical coefficients presented in Table 4 and Table 5, the slight volatility of 1 unit results in a significant change in LEP. Although LEP in various countries generally fluctuated from 1995 to 2015, under the joint effects of multiple variables, the impact of energy consumption and pollutant emissions on public health in high-income industrial countries was still obvious.

Further analysis based on empirical results in terms of independent variables reveals the following facts. (1) The contribution of energy consumption and pollutant emissions to LEP indicates that, for powerful industrialized countries, developing and organizing cleaner production is more beneficial than restricting the scale of energy consumption. (2) Empirical results confirm the positive effect of energy consumption in previous studies, i.e., improving residents’ life quality and social welfare. Thus, this paper argues that although natural disasters, extreme weather, and heavy pollution caused by large amounts of energy consumption will endanger public health in some regions, it should not deny the overall positive effect of such consumption in one country, especially in the case where GDPEU has presented a sustained upward trend. In China, for instance, the fog and haze appearing in large cities in recent years has increased the concentration of inhalable particulate matter and the incidence of respiratory diseases, but this pollutant has not spread widely or consistently; thus, its damage to public health has not reached a severe level. (3) According to the explanations of the World Bank and previous research [41], we argue that the increase of energy use efficiency will help decrease the scale of pollutant emissions and thus contribute to the improvement of public health; however, this should be based on a decrease or unchanged level of total energy consumption. In fact, with the exceptions of Japan, Germany, UK, and Belgium, total energy use has experienced an increasing trend in all countries, which may weaken the positive effect of increased energy use efficiency in the long-term. This case highlights that the positive impact of EUPC on LEP does not indicate that governments should continue to encourage fossil energy consumption, but only that the positive effect of such consumption at the current phase outweighs the negative effect. In the context of general increases of both GDPEU and the total population in our sample countries, it is necessary to control fossil energy consumption to avoid the reversal of its positive effect. (4) The negative effects of energy consumption and its consequences are usually under the control of governments of high-and middle-income countries that have the capability to prevent such effects in the short-term, even if they have experienced severe pollution events, e.g., seasonal haze. In other words, for developed countries, there are few energy use activities that constantly endanger public health currently. Thus, even if EUPC and the emissions of major pollutants are on the rise, there is no significant negative effect on public health. Furthermore, the contribution of energy use to LEP suggests that it will not significantly decrease EUPC in the short-term, thus, cleaner production is a key approach to improve public health, which expands the previous research in terms of social value of energy consumption.

Second, integrating the definition and measuring unit of our variables, this paper concludes that the scale of healthcare resources, e.g., government medical input and residents’ healthcare willingness, has the most positive impact on public health compared with other determinants. Such resources can be also seen as a key driving force for LEP in the face of increased energy consumption and pollutant emissions in various countries. In the context of large-scale energy consumption, the role of healthcare can be described as follows. Although new energy and cleaner production technologies are broadly supported by industrialized countries, the demand for these sustainable resources in these countries is always greater than their supply. Thus, in order to ensure the expected economic growth, we will remain dependent on fossil fuels in the future. At the same time, energy use efficiency may not be improved rapidly, which will trigger an increasing trend of the scale of pollutant emissions.

Third, education investment also significantly contributes to the increase in LEP. Compared with previous studies [37,42], we find that the positive significance of education for residents is far more significant than the public general expectation, and its marginal value has presented an increasing state [37,42]. This is mainly due to residents beginning to receive education at a younger age, and this design plays a decisive effect in establishing their social cognition and code of conduct. Compared with physical healthcare resources, education also plays an effect in improving mental health. Specifically, the level of education generally positively affects the employment rate and residents’ income. Under the premise of a perfect national medical security system, a more advanced education system will lead residents to pursue healthier lifestyles. The distribution of our data indicates that the average proportion of education as a share of GNI in sample countries was 4.32% during the period 1995 to 2015, with an upward trend, which suggests governments of powerful industrialized countries are focusing more on education input. Our empirical results indicate that, for this type of country, education and healthcare resources jointly constitute the key driving force of improving public health. Thus, even if energy consumption and pollutant emissions both present upward trends, their negative impacts on public health will weaken due to improved social security.

Fourth, in addition to the above analysis in terms of the factors affecting LEP, the effect of PUA also should be noted. Empirical results indicate that the impact of PUA on LEP is unstable. Overall, population concentration in large cities in powerful industrialized countries did not considerably increase LEP, which further suggests a positive effect of regional disparities in infrastructure construction narrowing in individual countries. In other words, even if residents tend to migrate to large cities, the public health in other regions will not be severely endangered by a lack of migration.

In the next section, we further analyze whether the levels of countries’ energy endowments and residents’ income can determine the impact of energy consumption and pollutant emissions on LEP, thus presenting how their own energy and economy scales will lead to heterogeneity among countries with different features.

## 4. Heterogeneity Analysis

### 4.1. Heterogeneity Analysis of Net Energy Exports and Importers

This study used “energy imports, net (% of energy use)” published by the World Bank as the basis for identifying net energy exporters (12 countries with an average proportion of net exports of 87.61 during 1995 to 2015) and importers (21 countries with an average proportion of net imports of 53.80 during 1995 to 2015). We find that both net exporters and importers involve high-income and low-and middle-income countries. Accordingly, the heterogeneity test will eliminate the potential impact of economic growth and, thus, more authentically present the decisive effect of energy endowment. Table 6 and Table 7 indicate heterogeneity results for these two types of countries; due to limited space, they only present the effect of independent variables (all control variables were actually included in the test process) and results of unit root and cointegration tests that have a statistical significance.

Through comparing Table 6 and Table 7, both EUPC and GDPEU significantly improve LEP in these two types of countries, and their regression coefficients are close to the results of the full data test, which further indicates the general positive effect of improving energy use efficiency. However, one issue to be noted is that pollutant emissions in these two types of countries negatively affects LEP, but only net importers produce a statistically significant effect. The average GHGPC, PM10, and LEP in net importers are 13.76, 63.52, and 72.14, respectively, while net exporter values are 9.67, 40.57, and 77.60, respectively. Potential reasons are as follows. First, net exporters may have a weaker control over the energy use scale and efficiency as their merits in energy endowment, which will lead to a higher EUPC and lower GDPEU, thus triggering the subsequent large-scale pollutant emissions. By contrast, net importers’ LEP is more sensitive to the marginal volatility of pollutant emissions, which may be caused by increasing energy importing costs as a result of the rise of pollutant emissions scale, thus, in turn leading to more efficient energy consumption and environmental governance activities in net importers. On the other hand, the average LEP of net exporters from 1995 to 2015 was much lower than that of net importers. Based on previous studies, we argue that aged social groups may be more sensitive to environmental quality and, as a result, a minor rise in negative environmental factors will trigger significant health damage in net importers. Integrating empirical test results in Table 4 and Table 5, this paper argues that there is no obvious heterogeneity in terms of the impact of energy consumption and pollutant emissions on LEP in these two types of countries.

### 4.2. Heterogeneity Analysis of High-Income and Low-and Middle-Income Countries

The World Bank has divided countries into high-income and low- and middle-income countries, and accordingly, our sample covers 21 high-income countries with average GDP per capita of $29,493.22 in the period 1995 to 2015, and 12 low- and middle-income countries with average GDP per capita of $5272.21. Similarly, these two types of countries include net energy exporters and importers; thus empirical results will exclude the potential impact of energy endowment and focus more on the decisive effect of economy scale. Same before, Table 8 and Table 9 only present the effect of independent variables, and the statistically significant results of unit root and cointegration tests. 

Through comparing Table 8 and Table 9, it can be found that the impact of EUPC and GDPEU on LEP in these two types of countries is consistent with the trend of the full data. Considering the large gap in the average LEP between these two types of countries (78.65 years in high-income countries and 70.31 years in low- and middle-income countries), EUPC and GDPEU in low- and middle-income countries still have remarkable potential for sustainably creating social welfare. Overall, progress in many socio-economic factors in low- and middle-income countries needs to be supported by energy consumption, which can still create common benefits for the majority of social groups. Given that both their average EUPC (1892.23 kg of oil equivalent) and GDPEU (8.76 PPP$ per kg of oil equivalent) are lower than those in high-income countries (4149.28 and 9.22, respectively), we predict that these two indicators will continue to rise in low- and middle-income countries, and governments may also present a willingness to constantly invest in such a consumption pattern.

With respect to effects of pollutant emissions, both GHGPC and PM10 in high-income countries significantly hindered the rise of LEP, but only PM10 in low- and middle-income countries produced a negative effect. A potential reason is as follows. PM10 in low-and middle-income countries (64.20 mcg/m^3^) are almost twice as high as in high-income countries (36.60), but their GHGPC is at a lower level (7.84 metric ton of CO_2_ equivalent), and also far below that in high-income countries (13.34). This case indicates that the production process in low- and middle-income countries requires greater attention to control the emission of solid particulate matter. Accordingly, we conclude that an important premise for the significant impact of pollutant emissions on public health may be that their emissions need to reach a certain level. That is, small-scale emission may not significantly affect public health. Overall, the impacts of pollutant emissions on LEP present a significant disparity between high-income and low- and middle-income countries.

## 5. Discussions and Implications

### 5.1. Discussions

Public health has attracted global attention of scholars, who have carried out valuable and meaningful research. These studies have made a significant contribution to the improvement of global public health. However, due to different sources of research data, the influence of fossil energy consumption and pollutant emissions on public health remains controversial. Furthermore, few studies have focused on the analysis of the impact factors of public health at the global level; particularly lacking is differential analysis of public health factors in different countries. This study explored the effect of energy consumption and pollutant emissions on public health in 33 high fossil energy consumption countries, and further determined the heterogeneity of influences on public health from the perspective of energy import–export scales and national income levels. The present study raised a number of issues for discussion.

First, several previous studies reveled that fossil fuel consumption has raised a range of environmental problems, such as global warming and environmental degradation, which affect public health [21,43,44]. However, the results of this paper show that per capita energy usage and energy consumption per unit of GDP have a significant positive relationship with public health. The most likely causes are that appropriate energy consumption and energy efficiency contribute to the improvement of residents’ quality of life and social welfare, which significantly improve public health [19,23]. Although severe pollution from natural disasters, extreme weather, and large energy consumption in some sample countries do affect the public health of some people in the short-term, it is undeniable that energy consumption plays a positive role in the overall development of the country.

Consistent with previous studies [21,43], results of this paper confirmed that energy efficiency can lead to the reduction of the scale of pollutant emissions, and thus contribute to the improvement of public health. However, the premise of this improvement is a decline or unchanged amount of energy consumption. In fact, with the exception of Japan, Germany, the United Kingdom and Belgium, total energy consumption is on the rise, which in the long run may weaken the positive effects of increased energy use. Therefore, our findings do not support the conclusion that the consumption of fossil fuels has been actively reduced [44,45]. The positive link between energy use per capita and life expectancy does not suggest that the government should continue to encourage energy consumption, but only indicates that the positive effects of fossil energy consumption outweigh the negative effects at the current stage. 

Second, our results showed that both PM10 and per capita greenhouse gas emissions were significantly negatively correlated with expected life expectancy. These findings that pollutant emissions cause negative impact on public health are similar to the results proposed by [2,46,47]. In contrast to [48], this paper does not provide empirical evidence for the conclusion that greenhouse gases have contributed to the human development index. In addition, the results show that the regression coefficient of PM10 emission concentration is small, but the negative influence on public health should not be ignored based on their measurement units. The reason is that the state of co-formation in the sample countries is that for every increase in PM10 emission concentrations of 1 microgram per cubic meter, life expectancy decreased by 0.0671 years. Although these coefficients are only theoretical contributions, a slight change in 1 unit has led to a more pronounced change in life expectancy compared to the overall numerical distribution of the respective variables in the sample countries. 

Third, the results of heterogeneity analysis showed that per capita greenhouse gas emissions significantly decrease the life expectancy in high-income countries compared to middle-income countries, but that PM10 emissions only have a significant negative effect on the life expectancy in middle-income countries. Our findings are a response to the suggestions of [1,49] that their research should be carried out under the premise of considering the heterogeneity of countries. The impact of pollutant emissions on life expectancy shows significant differences in high- and middle-income countries. Based on this finding, we conclude that air pollutants negatively affect public health only if the pollution emission scale reaches a certain level. These findings further enrich the understanding of the link between environmental pollution and public health [33,47], and provide new empirical evidence for the differences in the impact of environmental pollution on public health. 

### 5.2. Policy Implications

Our findings provide some implications for policy makers, as follows: First, the core entity that both expands the industrial economy and produces pollutant emissions are manufacturing firms, and the effect of emission reduction policy ultimately depends on whether such firms will effectively implement these policies. Subject to the reliance on fossil energy, low- and middle-income countries are generally facing a dilemma regarding cleaner production. One valuable practice for reducing emissions developed in Western countries is industrial symbiosis, whose production process represents the emission reduction mechanism of manufacturing firms. However, this practice has not been broadly developed in low- and middle-income industrialized countries, because industrial sectors remain in a relatively isolated state, which makes it difficult to centrally manage emitted pollutants. To address this problem, governments should establish a coordinated emission reduction mode between government research institutions and manufacturing firms, and then provide financing convenience and financial subsidies, as well as advanced technologies to encourage the spontaneous cooperation among firms [50]. This would not only compensate manufacturing firms’ loss due to their emission reduction actions, but also motivate them to invest more resources to develop new emission reduction projects with the aim to push forward the national environmental governance.

Second, drawing on the advanced experience in Western countries, the low- and middle-income countries should motivate manufacturing firms to develop green product innovation and advocate social groups’ consumption of green products. One approach broadly popularized in Western countries is green governmental procurement, which previous studies have shown has greater potential in developing countries as a key approach to restrict pollutant emissions, and thus create economic performance at the industrial level. Green governmental procurement will help optimize manufacturing firms’ production structures and raise public concerns regarding green products, which will significantly contribute to the decrease of pollutant emissions.

Third, the empirical results do not suggest that governments should continue to encourage fossil energy consumption. In the long-term, its replacement with new energy and significant improvements of energy efficiency will become an inevitable trend. At this stage, regional or seasonal diseases related to fossil energy consumption suggest that the governments should take controlling pollution sources as the primary approach to improve public health when they are unable to significantly reduce fossil energy consumption.

## 6. Conclusions

This paper explores the impact of fossil energy consumption and pollutant emissions on public health considering the population distribution, social resource allocation, and social factors of medical and health resources. Furthermore, 33 countries are further subdivided into net energy exports and net importers, high- and low- and middle-income countries, and this study identifies whether there are significant differences in the effect of fossil energy consumption and environmental pollution on public health.

The results show that fossil energy consumption and energy efficiency are positively associated with public health, and, on the contrary, pollutant emissions significantly and negatively affect public health. In addition, this paper also reveals that there are significant differences in the intensity and manner of the impact of per capita greenhouse gases on public health in different countries, but there is only one difference in the intensity of the impact of PM10, fossil energy consumption, and energy efficiency on public health in different countries. The present study suggets that the government should improve the civil awareness of environmental protection and health problems via controlling fossil energy consumption and pollutant emissions, especially in low- and middle-income countries and net energy importers.

Whether for one country or more countries, or one region or many regions, the difficulty in exploring the link between fossil energy consumption, pollutant emissions, and public health lies in how to accurately identify the most important factors affecting public health in a complex economic system. In particular, recognition of the differences between different countries in the understanding of the links between energy, environment, and public health, plays a key role in promoting the healthy development of society.

There are three limitations of this paper. First, although we reveal that the life expectancy of population can describe the level of public health in one country, and that this indicator will help accurately assess the social effect of energy consumption, the health problems caused by energy consumption and its consequence are still concentrated in the incidence of some diseases, and these diseases mainly occur in regions with concentrated manufacturing industries, dense populations, and developed economies. This paper does not further refine the typical energy consumption regions of our sample countries, which may mean empirical results do not represent the social hazards from energy consumption occurring in heavy-pollution regions. Second, this paper finds that fossil energy stock, energy import scale, and domestic fossil energy consumption preference jointly determine the quality of the national industrial economy and residents’ lifestyles, but the disparity of influencing factors on public health between energy abundant (e.g., Saudi Arabia, Iran, and Algeria) and non-energy abundant countries (e.g., Japan, South Korea, and Singapore) need to be further investigated. Third, although we selected variables as comprehensively as possible, some were not being selected due to missing data, e.g., the proportion of poor people (we expect that larger-scale poverty can hinder the increase of the life expectancy of the population), even if these variables may not alter the current empirical results. Therefore, data collection should be further expanded in future research.

## Figures and Tables

**Figure 1 ijerph-16-04678-f001:**
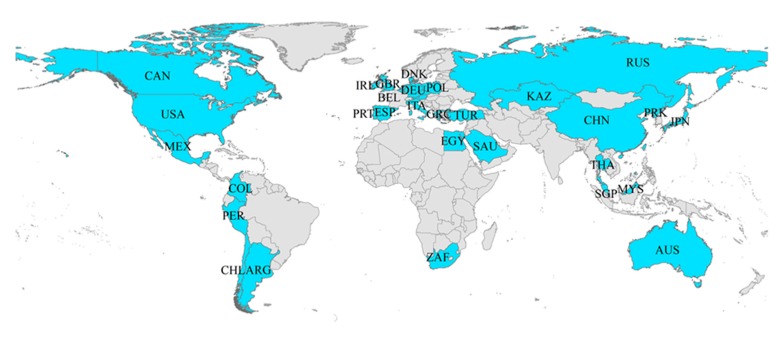
The geographical location of samples in this paper.

**Table 1 ijerph-16-04678-t001:** Relevant research on the link between pollutant emissions and public health.

Author (s)	Key Words	Sample Source	Results
Pascal et al. (2013) [12]	Health, pollutant emissions	12 European countries	The largest health burden was attributable to the effect of chronic exposure to PM2.5.
Kelly and Fussell [9]	Pollutant emissions, public health	Death rate	Air pollution is negatively related with public health.
Tang and Nagashima [13]	PM2.5, human health	10 regions in the world	PM2.5 seriously affects public health.
Bari and Kindzierski [14]	Source apportionment, potential risk for human health	Edmonton, Canada	PM2.5 can generate potential risk for public health.
Emife et al. [15]	Pollutant emissions, health outcome	Nigeria	Pollutant emissions has a negative effect on health outcome in Nigeria.
Lu et al. [16]	Environmental pollution, public health	30 provinces of China	Environmental pollution has a negative effect on public health, but education and medical conditions contribute to improving public health.
Sharma et al. [7]	Air pollution, public health	Delhi in India	Air pollution are also contributors to the causation of such diseases
Torres et al. [11]	Air pollution, public health	Portugal	With the fall of air quality, and the mortality rate also declined. There is no a significant link between air pollution and public health in Alentejo and Algarve.
Yang et al. [17]	Air pollution exposure, household healthcare expenditure	30 provinces of China	Air pollution leads to the increase in household healthcare expenditure and substantial adverse impacts on public health.
Khan et al. [8]	Pollutant emissions, psychiatric disorders	US and Denmark	Environmental pollution crease risk of psychiatric disorders.
Xu et al. [1]	Industrial air pollution, public health	30 provinces of China	Air pollution is negatively associated with public health in almost all of regions in China.

**Table 2 ijerph-16-04678-t002:** Relevant research on fossil energy consumption and public health.

Author (s)	Theme	Sample Source	Results
Ozturk [28]	Energy consumption, air quality, countries’ health.	Australia, Austria, Belarus, Belgium, Bulgaria, Canada	Fossil Energy consumption plays an important effect in exacerbating climate change, which further affects public health.
Asikainen et al. [29]	Greenhouse gas emissions, PM2.5 and health effects	Kuopio in Finland	The reduction of fossil energy consumption and the improvement of energy efficiency can reduce environmental pollution and benefit public health
Erickson and Jennings [22]	Energy, air quality, climate change, health	USA	There will be human health benefits from reducing fossil energy consumption emissions in all parts of the world.
Perera [10]	Child health, air pollution, climate change, fossil fuel combustion	NA	Air pollution and climate change from fossil fuel combustion can affect children health.
Spiegel and Brown [20]	Coal extraction on miners, local populations.	NA	Carbon emissions and harmful air particles from fossil energy consumption have a direct impact on the health of surrounding communities.
Chandio et al. [19]	Energy growth, environment quality	Pakistan	The increase in economic growth and electricity consumption in the agriculture sector degrades environmental quality in Pakistan

**Table 3 ijerph-16-04678-t003:** Descriptive statistics.

Variables	Unit	Min	Max	Mean	S.D.
PD	people per sq. km of land area	2.35	7806.77	317.77	1113.83
PUA	%	4.28	100.00	30.42	18.88
UEM	%	0.70	27.47	7.91	5.02
EE	%	1.70	8.07	4.32	1.29
IHF	%	53.2	100.00	92.02	11.51
PHE	%	0.89	10.12	4.81	2.16
PCHE	%	2.44	17.46	7.31	2.73
EUPC	kg of oil equivalent	432.03	8441.18	3328.53	1849.63
GDPEU	PPP $ per kg of oil equivalent	2.51	21.5	9.05	3.28
GHGPC	metric ton of CO_2_ equivalent	1.66	58.37	11.34	7.76
PM10	microgram/m^3^	12.13	173.75	46.64	29.30
LEP	Year	51.56	83.84	75.62	5.60

Note: PPP means the purchasing power parity.

**Table 4 ijerph-16-04678-t004:** Empirical results through the fixed effect model.

Variables	Model (1)	Model (2)
Constant	49.2995 *** (4.2340)	57.9005 *** (6.2586)
PD	0.0002 (0.0003)	0.0015 *** (0.0003)
PUA	−0.0293 (0.0558)	0.0955 ** (0.0615)
UEM	0.0266 * (0.0435)	−0.0940 *** (0.0540)
EE	0.3086 *** (0.2212)	0.3158 *** (0.2353)
IHF	0.1019 *** (0.0487)	0.1298 *** (0.0582)
PHE	0.4578 *** (0.1513)	0.4370 *** (0.1709)
PCHE	0.3003 *** (0.0786)	0.4244 *** (0.1234)
EUPC	0.0016 *** (0.0003)	
GDPEU	0.7273 *** (0.0937)	
GHGPC		−0.0269 (0.0317)
PM10		−0.0671 *** (0.0288)
N	693	693
Unit Root (lag: 1, Levin, Lin, and Chu test)	−233.9640 *** (0.0000)	−233.972 *** (0.0000)
Unit Root (lag: 1, Breitung t-stat)	−3.2873 *** (0.0005)	−3.4142 *** (0.0003)
Unit Root (lag: 1, Im, Pesaran and Shin W-stat)	−78.8627 *** (0.0000)	−79.4594 *** (0.0000)
Unit Root (lag: 1, ADF—Fisher chi-square)	726.145 *** (0.0000)	729.075 *** (0.0000)
Unit Root (lag: 1, PP—Fisher chi-square)	802.543 *** (0.0000)	823.008 *** (0.0000)
Cointegration Test (lag: 2, Kao Residual Cointegration Test)	765.166 ** (0.0203)	758.9713 ** (0.0155)
Cointegration Test (lag: 2, Johansen Fisher Panel Cointegration Test)	1543 *** (0.0000)	1679 *** (0.0000)
F-value	204.28 ***	127.30 ***
Adjusted R^2^	0.7385	0.6377

Note: ***, **, and * are statistically significant at 1%, 5% and 10%, respectively, with the same as Table 5, Table 6, Table 7, Table 8 and Table 9. Values in the parentheses in Model (1) and Model (2) corresponding to each variable are standard error of clustering at the national level, and values in the parentheses corresponding to both unit root and cointegration tests are *p*-values.

**Table 5 ijerph-16-04678-t005:** The robustness test.

Variables	Model (1)	Model (2)
Constant	49.8020 *** (4.0388)	56.0640 *** (6.1732)
PD	1.12 × 10^−7^ (1.42 × 10^−7^)	7.40 × 10^−7^ *** (1.26 × 10^−7^)
PUA	−1.84 × 10^−5^ (2.82×10^−5^)	3.90 × 10^−5^ ** (3.01 × 10^−5^)
UEM	1.29 × 10^−5^ * (2.10 × 10^−5^)	−4.33 × 10^−5^ *** (2.64 × 10^−5^)
EE	0.0002 *** (0.0001)	0.0002 *** (0.0001)
IHF	0.0001 *** (2.35 × 10^−5^)	0.0001 *** (2.85 × 10^−5^)
PHE	0.0002 *** (0.0001)	0.0002 *** (0.0001)
PCHE	0.0001 *** (3.79 × 10^−5^)	0.0002 *** (0.0001)
EUPC	7.53 × 10^−7^ *** (1.42 × 10^−7^)	
GDPEU	0.0003 *** (4.48 × 10^−5^)	
GHGPC		−1.16 × 10^−5^ (1.50 × 10^−5^)
PM10		−2.89 × 10^−5^ *** (1.38 × 10^−5^)
N	693	693
F-value	211.91 ***	133.34 ***
Adjusted R^2^	0.7458	0.6487

**Table 6 ijerph-16-04678-t006:** Results of net energy exporters.

Variables	Model (1)	Model (2)
EUPC	0.0012 *** (0.0002)	
GDPEU	0.7019 *** (0.0965)	
GHGPC		−0.0169 (0.0276)
PM10		−0.0605 *** (0.0077)
N	252	252
Unit Root (lag: 1, Levin, Lin, and Chu test)	−31.4518 *** (0.0000)	−31.4141 *** (0.0000)
Unit Root (lag: 1, Im, Pesaran and Shin W-stat)	−109.229 *** (0.0000)	−108.221 *** (0.0000)
Unit Root (lag: 1, ADF—Fisher chi-square)	408.306 *** (0.0000)	401.409 *** (0.0000)
Unit Root (lag: 1, PP—Fisher chi-square)	521.995 *** (0.0000)	522.439 *** (0.0000)
Cointegration Test (lag: 2, Johansen Fisher Panel Cointegration Test)	547.6 *** (0.0000)	619 *** (0.0000)
F-value	41.35 ***	36.18 ***
Adjusted R^2^	0.6170	0.5850

Note: Net exporters are Canada, Russia, Australia, Mexico, Saudi Arabia, Argentina, Egypt, South Africa, Denmark, Malaysia, Colombia, and Kazakhstan. Other countries are net importers.

**Table 7 ijerph-16-04678-t007:** Results of net energy importers.

Variables	Model (1)	Model (2)
EUPC	0.0018 *** (0.0004)	
GDPEU	0.6915 *** (0.1302)	
GHGPC		−0.1670 *** (0.1089)
PM10		−0.1144 *** (0.04346)
N	441	441
Unit Root (lag: 1, ADF—Fisher chi-square)	295.467 ** (0.0311)	295.802 ** (0.0302)
Unit Root (lag: 1, PP—Fisher chi-square)	300.407 ** (0.0196)	291.783 ** (0.0431)
Cointegration Test (lag: 2, Kao Residual Cointegration Test)	1173.528 ** (0.0450)	1150.980 ** (0.0454)
Cointegration Test (lag: 2, Johansen Fisher Panel Cointegration Test)	995.5 *** (0.0000)	1060 *** (0.0000)
F-value	346.05 ***	222.66 ***
Adjusted R^2^	0.8834	0.8298

**Table 8 ijerph-16-04678-t008:** Results of high-income countries.

Variable	Model (1)	Model (2)
EUPC	0.0015 *** (0.0004)	
GDPEU	0.7350 *** (0.1331)	
GHGPC		−0.0484 *** (0.0402)
PM10		−0.0638 *** (0.0306)
N	441	441
Unit Root (lag: 1, Levin, Lin, and Chu test)	−7.5857 *** (0.0000)	−7.9228 *** (0.0000)
Unit Root (lag: 1, Im, Pesaran and Shin W-stat)	−16.063 *** (0.0000)	−16.2507 *** (0.0000)
Unit Root (lag: 1, ADF—Fisher chi-square)	749.407 *** (0.0000)	756.787 *** (0.0000)
Unit Root (lag: 1, PP—Fisher chi-square)	1792.87 *** (0.0000)	1716.78 *** (0.0000)
Cointegration Test (lag: 2, Kao Residual Cointegration Test)	1140.397 ** (0.0313)	1121.842 ** (0.0460)
Cointegration Test (lag: 2, Johansen Fisher Panel Cointegration Test)	1083 *** (0.0000)	1125 *** (0.0000)
F-value	314.38 ***	157.76 ***
Adjusted R^2^	0.8732	0.7755

Note: High-income countries are United States, Japan, Germany, United Kingdom, Italy, Canada, Korea, Australia, Spain, Netherlands, Saudi Arabia, Poland, Belgium, Austria, Denmark, Singapore, Ireland, Chile, Portugal, Greece, and Czech Republic. Other countries belong to the low- and middle-income group.

**Table 9 ijerph-16-04678-t009:** Results of low- and middle-income countries.

	Model (1)	Model (2)
EUPC	0.0014 *** (0.0003)	
GDPEU	0.6254 *** (0.2476)	
GHGPC		0.0945 (0.0950)
PM10		−0.0879 *** (0.0177)
N	252	252
Unit Root (lag: 1, Levin, Lin, and Chu test)	−28.6721 *** (0.0000)	−28.6700 *** (0.0000)
Unit Root (lag: 1, Im, Pesaran and Shin W-stat)	−109.585 *** (0.0000)	−109.294 *** (0.0000)
Unit Root (lag: 1, ADF—Fisher chi-square)	426.568 *** (0.0000)	420.376 *** (0.0000)
Unit Root (lag: 1, PP—Fisher chi-square)	557.419 *** (0.0000)	553.373 *** (0.0000)
Cointegration Test (lag: 2, Johansen Fisher Panel Cointegration Test)	460.5 *** (0.0000)	554.2 *** (0.0000)
F-value	43.43 **	40.41 ***
Adjusted R^2^	0.6285	0.6116

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
