# Peer review of "How Energy Consumption and Pollutant Emissions Affect the Disparity of Public Health in Countries with High Fossil Energy Consumption"

_ijerph, 2019, doi:10.3390/ijerph16234678_

Round 1

Reviewer 1 Report

Recommendation: Major Revision

I have carefully considered and read the manuscript entitled “How Energy Consumption and Pollutants Emission Affect the Disparity of Public Health in Countries with High Fossil Energy Consumption” and have the following observations:

The manuscript discussed the issues on public health are being broadly focused on worldwide, and recent research has begun to investigate the antecedents of public health in terms of their disparities in different regions. Based on the data from 33 countries with both higher GDP and the proportion of fossil energy consumption from 1995 to 2015, this paper identifies these antecedents from the perspective of energy consumption and pollutant emission. Our empirical results indicate that the total consumption of fossil energy is beneficial to the life expectancy of the population (LEP), and the positive effect of energy use efficiency on LEP is more significant.

Major comments and Suggestions for Authors:

This paper is not enough clarity about the aims and objectives, so please do it clarify. There are some errors in spelling, and some more clarifications and improvements are needed for reconsidering it for publication in the International Journal of Environmental Research and Public Health.

In addition to the above, I have a few points for the authors to consider them before the publication of this work:

The abstract should check thoroughly and compose it, with a summative mode without articles and spelling errors. Please highlight your contribution and novelty of this manuscript with accuracy in the introduction part before the arrangement description. Furthermore, the objectives of your study should elaborate clearly there in the introduction part. The literature should add more relevant studies e.g. (2019) to grab and display more contemporary literature critically. Please update your literature with few latest studies if it suitable:

Check whether the following reference are relevant and may be cited:

Rauf, A.; Liu, X.; Amin, W.; Ozturk, I.; Rehman, O.U.; Sarwar, S. Energy and Ecological Sustainability: Challenges and Panoramas in Belt and Road Initiative Countries. Sustainability 2018, 10, 2743.

Chandio, A.A.; Jiang, Y.; Rauf, A.; Mirani, A.A.; Shar, R.U.; Ahmad, F.; Shehzad, K. Does Energy-Growth and Environment Quality Matter for Agriculture Sector in Pakistan or not? An Application of Cointegration Approach. Energies 2019, 12, 1879.

Recheck the references and their style are according to the journal requirements, and in-text and end-text should be the same and vice versa. In the Methodology part, please modify it based on your cross-dependence test further e.g. (CD test first, then either 1st or 2nd generation unit root test, then based on unit root go for cointegration test and then move ahead for (DOLS, FMOLS, CCEMG, PMG) in comparison way In the result and discussion section, some associated literature must be added to compare and contrast the key findings with the existing studies. Furthermore, Study limitations should be moved from the result and discussion part to final conclusion part but not in the result and discussion part. The conclusion should be based on your results and discussion. So, do consider it and improve it based on the logic of your results. The conclusion does not properly describe as it was needful, hence please provide expansion in your conclusion-based estimations and provide some recommendations and policy implications accordingly. The acronyms should be defined at first appearance in the manuscript and then must be consistently used throughout the manuscript. Furthermore, the manuscript must be checked form typo errors and spelling checks.

Reviewer 2 Report

The paper entitled “How Energy Consumption and Pollutants Emission 3 Affect the Disparity of Public Health in Countries 4 with High Fossil Energy Consumption”

There some more comments on the manuscript.

Abstract:

Please write clear aim of the study.

Add key conclusions form each section of the results and discussion.

Introduction:

Authors have stated good literature study in this paper, but they not interesting in this format. I would suggest to cite more recent studies in this areas and make a table for all of them as showing a table is much clear for readers.

In addition. The aim of the study isn’t clear and need to be re-written the paragraph and compare with previous studies and identify gap research.

Research design:

Please add and make a world map for all cities which used in this study.

Please move some materials in this section to the supplementary materials as they are too much in the present form.

Result and discussion section:

Very weak statement without supporting them by previous studies. Please make sure cite the relevant studies and compare with your findings.

In the current form, it’s not interested to reader. Add some figures to show your results. Too much Tables without pay attention to show them in scientific way.

Conclusions:

Please re-write and change the structure to bulling points. You can write brief conclusion for each section of the results and discussion. Please add the gap study and remain research question in this filed. I would suggest write the shortcomings of this study and future work in this research area.

Round 2

Reviewer 1 Report

Paper is hugely improved and expressed as a case in an appropriate way. The current paper has an acceptable language style for the readership of IJERPH. Research implications are properly written and, so far this paper is accepted in recent format for final publication in IJERPH. The results are designated appropriately and the final conclusion is completely based on retrieved estimations. Hence, all sections are synchronizing, coherent and well sectioned. The methodology is well documented, work is intellectually well written and appropriately important methods are utilized with core insight to provide some strong policy implications. The paper demonstrated adequate insights into the related literature appropriately well written in the current manuscript that will meet the standards of IJERPH readership. In my own opinion, no more significant work is ignored in this current manuscript. Hence, I recommend for publication with strong acceptance.

Reviewer 2 Report

Thank you for authors. They have addressed all my comments.